# Analysis of the Factors Influencing Body Weight Variation in Hanwoo Steers Using an Automated Weighing System

**DOI:** 10.3390/ani10081270

**Published:** 2020-07-25

**Authors:** Hyunjin Cho, Seoyoung Jeon, Mingyung Lee, Kyewon Kang, Hamin Kang, Eunkyu Park, Minkook Kim, Seokman Hong, Seongwon Seo

**Affiliations:** 1Division of Animal and Dairy Sciences, Chungnam National University, Daejeon 34134, Korea; chohyunjin0927@gmail.com (H.C.); seoyoung203@gmail.com (S.J.); mingyung1203@gmail.com (M.L.); kangkyewon26@gmail.com (K.K.); gkals0339@gmail.com (H.K.); 2Woosung Feed Co., Ltd., Daejeon 34379, Korea; ekpark@wsfeed.co.kr (E.P.); mkkim@wsfeed.co.kr (M.K.); smhong@wsfeed.co.kr (S.H.)

**Keywords:** analysis of variance components, automated weighing system, Hanwoo

## Abstract

**Simple Summary:**

The body weight (BW) of animals is an important indicator of their physiological status and productivity. The BW of animals varies from day to day and even within a day due to various factors. However, these variations have not been fully tested because it is challenging to measure the BW of animals repeatedly at various time points. This study used an automated weighing scale (AWS) to overcome these difficulties and generated a large number of BW measurements. We found that differences between individual animals had the greatest impact on BW deviations in Hanwoo steers. Additionally, it was found that changes in the BW of Hanwoo steers during the day were influenced by feeding patterns. To the best of our knowledge, this is the first study to report the diurnal pattern of changes in the BW of Hanwoo steers. Our results suggest that variations in individual animals and their feeding patterns need to be considered when applying precision-farming technologies with real-time BW measurements in cattle.

**Abstract:**

This study aimed to determine the factors affecting the body weight (BW) of Hanwoo steers by collecting a large number of BW measurements using an automated weighing system (AWS). The BW of 12 Hanwoo steers was measured automatically using an AWS for seven days each month over three months. On the fourth day of the BW measurement each month, an additional BW measurement was conducted manually. After removing the outliers of BW records, the deviations between the AWS records (a) and manual weighing records (b) were analyzed. BW measurement deviations (a − b) were significantly (*p* < 0.05) affected by month, day and the time within a day as well as the individual animal factor; however, unexplained random variations had the greatest impact (70.4%). Excluding unexplained random variations, the difference between individual steers was the most influential (80.1%). During the day, the BW of Hanwoo steers increased before feed offerings and significantly decreased immediately after (*p* < 0.05), despite the constant availability of feeds in the feed bunk. These results suggest that there is a need to develop pattern recognition algorithms that consider variations in individual animals and their feeding patterns for the analysis of BW changes in animals.

## 1. Introduction

The body weight (BW) of animals represents their physiological status and growth rate and is an important basis on which animal management strategies are decided. In the field, the BW of Hanwoo steers is measured once every few months. However, measuring BW once over the course of a few months is not reliable because the BW of animals can fluctuate considerably from day to day due to various factors, including environmental temperature, age and size of animals [1], as well as feed and water intake [2]. Despite this, repeated BW measurements of animals have been avoided because they can cause stress and sometimes harm steers.

The recent development of an automated weighing system (AWS) has helped overcome this problem. The AWS can continuously monitor changes in the BW of an animal more objectively and with less labor than traditional manual BW measurement [3]. A walk-over scale, for example, allows repeated measurement of BW of animals without restraint while the animals are traversing the weighing platform before or after milking [4]. The automated milking system is often equipped with a static AWS that measures BW of cows during milking [5]. Alawneh et al. [6] stated that the AWS, which can measure BW frequently and does not stress the animals, has many advantages over traditional BW measurement methods, and it can be used as an indication of the animal’s physiological health. Due to these advantages, several studies have recently been conducted to apply AWS in the field. Pszczola et al. [5] conducted a study to increase the accuracy of BW measurement by repeatedly using an automated milking system equipped with a scale. Alawneh et al. [6] developed an algorithm to use the AWS for herd management in pasture-fed dairy cows. Dickinson et al. [4] indicated that the AWS could be used for confirming small changes in animal BW after removing outliers that are incorrectly recorded due to AWS malfunction or other factors.

However, to the best of our knowledge, no study has investigated the influence of the factors that can cause variation in the BW measurement of Hanwoo steers. In particular, the changes in BW of Hanwoo steers over the course of a day are not known. Therefore, this study was conducted to collect many BW measurements by repeatedly measuring BW of Hanwoo steers using an AWS and determine the causes of variation in the BW measurement of Hanwoo steers and the pattern of variation in BW of Hanwoo steers over the course of a day.

## 2. Materials and Methods

This study was conducted from February 2019 to April 2019. For each month, BW measurements were performed for seven days using an AWS (Dawoon Co., Incheon, Korea) connected to an automated concentrate feeder (ACF; Dawoon Co., Incheon, Korea). The experiment was conducted at the Center for Animal Science Research, Chungnam National University, Korea. The use of animals and the protocols for this experiment were reviewed and pre-approved by the Chungnam National University (CNU) Animal Research Ethics Committee (CNU- 01021).

Twelve 11-month-old Hanwoo steers were used in this experiment. The initial mean BW (±standard deviation, SD) of Hanwoo steers was 319 (±29.4) kg. Randomly selected steers were housed in a pen (10 × 10 m^2^) that had one ACF and four forage feed bunks. Each ACF and forage feed bunk was equipped with a real-time electronic individual feeding system that recognized each steer entering the feeder by sensing the radio-frequency identification (RFID) neck tag attached to each animal (Dawoon Co., Incheon, Korea). When a steer entered the ACF, regardless of feed offering, a real-time electronic system within ACF recorded the presence of the steer and measured BW. The forage was fed ad libitum twice a day at 07:00 and 17:30, and a commercial concentrate mix was fed through an ACF. Nutrient composition and amount of concentrate mix provided were determined according to the Korean feeding standards for Hanwoo steers (NIAS, 2017), aiming at an average daily gain (ADG) of 1 kg. Diet composition of the concentrate mix and the chemical composition of the experimental diets are described in Table 1 and Table 2. Each day was divided into four periods of 6 h; within each period, steers were able to consume up to one-fourth of the amount of daily allowable concentrate mix. If steers did not consume the amount of concentrate mix allowed during each period, they could consume the rest in the next period.

For each measurement period, we cleaned the manure in the space to install the AWS. We placed the scale, ensuring that it was stable and did not come in contact with any object, and then connected the plug to the ACF. The scale calibration was performed after every AWS installation to the ACF. We used 418 kg as the calibration weight for targeting 1000 kg, which is more than 1/3 of the target weight recommended by the product manufacturer. We repeatedly measured weights using the AWS by adding 20 kg weights up to 458 kg, and then decreasing the weight by 100 kg to 158 kg (i.e., a total of five different weight measurements), to evaluate whether the AWS measurements were accurate after calibration, and ensure that the deviation of each measurement was less than 10 kg.

For each measurement period, BW of the steers was measured for seven days after the AWS was installed. On the fourth day of BW measurement, BW was manually measured using an electronic weighing scale before the morning feeding. We confirmed that the ACF and AWS were connected correctly and that the AWS did not touch the wall to ensure that BW measurements were accurate. In addition, feces and urine on the AWS were removed once or twice a day during the BW measurement period.

After seven days of BW measurement using the AWS, outliers of the collected BW measurement records were removed. A reasonable value was set for each animal (average BW for seven days ± 10% of the average BW for seven days), and measurements outside the criteria were assumed to be outliers, which were removed until none were present. The BW measurement records were normalized for conducting statistical analysis after the outliers had been removed. The deviations between the BW records measured by AWS (a) and those measured manually by the static weighing scale (b) were calculated for each animal and measurement period. The deviation value (a − b) was defined as the BW measurement deviation, which was used for the statistical analysis. We considered the animal as a random effect and the measurement month, measurement day and measurement time within a day as fixed effects and the significant factors affecting BW measurement deviation. The measurement time within a day was expressed as three-hour time periods by dividing a day (i.e., 00:00–24:00) into eight three-hour zones. Dry matter intake (DMI) and initial BW of each period were analyzed using PROC MIXED of SAS (SAS Institute Inc., Cary, NC, USA). The significance of these variance components was analyzed using PROC MIXED of SAS to calculate the level of variation caused by each factor. In this analysis, all factors were treated as random effects. In addition, the least square means of the deviations for each time zone were calculated using the PROC GLIMMIX of SAS, and the differences between the time zones were analyzed.

## 3. Results

BW measured for seven consecutive days a month for three months using an AWS collected an average of 10.5 BW measurement records per day per animal. Consequently, 2656 BW records were collected during the experiment. Among these records, 33 records (1.2%) were removed as outliers.

The initial BW of each period and average DMI during the seven-day experimental period are described in Table 3. The BW of steers continued to increase as the period passed. The concentrate mix intake was significantly different (*p* < 0.001) for each period and was the largest in Period 3.

The overall mean BW measurement deviation was 9.7 kg. In this study, the BW measurement deviation was defined as the difference between the BW records measured by the AWS (a) and the BW records measured by the electronic weighing scale (b). In each measurement month, the BW measurement deviation was 14.5, 5.7 and 8.7 kg for the months of February, March and April, respectively, with a standard error of 0.75. The BW measurement deviation was significantly (*p* < 0.05) affected by all factors; however, unexplained random variations accounted for 70.4% and had the greatest impact. The analysis of the influence of each factor, excluding unexplained random variations, showed that the difference between animals was the most influential (80.1%), and the influences of the remaining factors, i.e., measurement month, measurement time within a day and measurement day, were 17.2%, 1.9% and 0.8%, respectively (Table 4).

When each day (i.e., 00:00–24:00) was divided into eight three-hour time zones, the BW measurement deviation of Hanwoo steers was stable from 21:00 to 06:00 without any marked change but increased from 06:00 to 09:00 (Figure 1). Thereafter, BW decreased sharply and was the lowest between 09:00 and 12:00, and then increased steadily and was the highest between 15:00 and 18:00 (*p* < 0.05).

## 4. Discussion

The study investigated the factors that can cause variation in the BW measurement of Hanwoo steers using an AWS. There were only a few outliers (i.e., 1.2% of the records) indicating that relatively stable BW measurement records could be collected using an AWS. However, the measurement was reliable only if the calibration was performed using appropriate methods every time an AWS was installed, and the proper maintenance protocol was performed during the BW measurement period.

Unexplained random variations accounted for 70.4% and had the greatest impact on BW measurement deviation. These variations may be due to the accumulation of excretion of feces and urine on AWS after the animal enters the AWS. However, because the amount of feces and urine accumulated on AWS was not measured in this study, the effect of feces and urine on BW measurement deviation could not be accurately determined. When the unexplained random variations were excluded, the factors that had the greatest influence on the BW measurement deviation were differences between animals (80.1%), followed by the measurement month (17.2%), measurement time within a day (1.9%) and measurement day (0.8%). Although all animals consumed the same feed, the differences between animals appeared to have the greatest influence on the BW measurement deviation because the DMI, growth performance, step or movement and other behaviors of each individual were different. It is considered that the measurement month affects the BW measurement deviation because the BW of animals increased as the period passed. In addition, since there is a difference in BW and DMI between the periods, the measurement month is considered to affect the BW measurement deviation.

To the best of our knowledge, this is the first study to show that the BW of Hanwoo steers has a pattern of variation over the course of a day. The changes in BW of Hanwoo steers during the day seem to be influenced by the pattern of feed offering. In this study, the concentrate mix was not fed directly, but by the ACF, and the daily intake set by the program was divided into four time periods of the day so that it could be steadily consumed. In contrast, forage was manually fed twice a day (07:00 and 17:30) to ensure that the steers consumed forage ad libitum. The BW of the animal increases after consumption of feed and water and decreases after the excretion of feces and urine. The consumption is greater than excretion during the increasing BW phase of Hanwoo steers, and excretion is higher than the intake in the decreasing phase of BW. The pattern of BW change in Hanwoo steers during the day showed an increase in BW before the feed offerings, and a significant decrease immediately after, despite the constant availability of feed in the feed bunk throughout the experimental period (Figure 1). This result shows that the supply of new feed affects the patterns of feed intake and excretion in Hanwoo steers.

The supply of new feed seems to stimulate both feed intake and excretion of feces and urine from Hanwoo steers. It has long been known that an increase in the number of feeding occasions generally leads to an increase in feed intake [7] because feeding induces feed intake [8]. Villettaz Robichaud et al. [9] also reported a pattern of increased feed intake at feeding ad libitum. In addition, the frequency of excretion of feces and urine increases with the increased feed intake [10]. Aland et al. [11] showed a pattern of increased excretion of feces and urine immediately after manual feeding. Vaughan et al. [12] found that the excretion of feces and urine during the day was highly correlated with visits to the feed bin, and the excretion of feces and urine had a constant pattern of increase after feeding. Pszczola et al. [5] investigated the pattern of BW change in dairy cows during the day and found the lowest BW before morning feeding and an increase in BW over the rest of the day. This may be because the feed was only fed once in the morning. Although this study did not directly investigate the excretion of feces and urine, the diurnal pattern of BW changes indicated that the supply of new feed, human activity related to feeding or both might induce animal feed intake, as well as promote excretion of feces and urine.

## 5. Conclusions

This study confirmed that the BW of Hanwoo steers varies depending on the feeding pattern. The results suggest that feeding patterns should be considered when developing algorithms to analyze BW changes in animals. The algorithm that analyzed BW changes in animals measured by the AWS is believed to be able to identify the pattern of BW changes for individual animals and indicate unusual conditions or health problems, thereby contributing to animal welfare and care. Adding the feeding pattern into the algorithm would increase the precision and power of detecting abnormal conditions in cattle through the use of the AWS. In the present study, however, variations in individual animals showed the greatest impact, accounting for 80.1% of the displacement (when unexplained random variables were excluded) and indicating the need to develop pattern recognition algorithms that consider the variations in individual animal.

## Figures and Tables

**Figure 1 animals-10-01270-f001:**
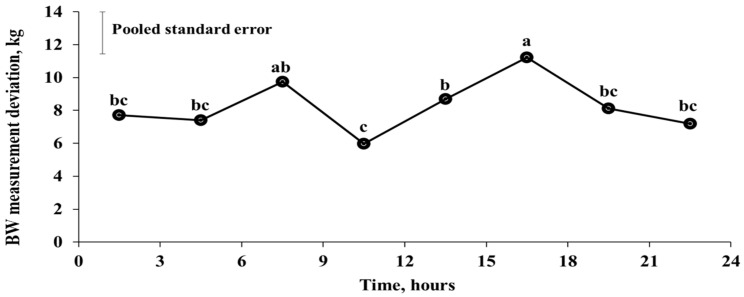
Daily variation BW measurement deviation with three-hour intervals. BW measurement deviation refers to the difference in BW between automated BW records and manual BW measurement. Means that do not have common superscripts (a–c) significantly differ (*p* < 0.05).

**Table 1 animals-10-01270-t001:** Analyzed chemical composition (g/kg DM or as stated) of the experimental diets.

Items ^1^	Concentrate	Forage
DM, g/kg as fed	868	914
OM	898	920
CP	195	75
SOLP	62	35
NDICP	26	8
ADICP	9	7
aNDF	295	656
ADF	127	428
ADL	31	56
Ether extract	35	11
Ash	102	80
Ca	17	3
P	7	1
K	13	24
Na	5	1
Cl	9	6
S	4	1
Mg	4	2
TDN	711	549
NEm, MJ/kg DM	6.8	5.2
NEg, MJ/kg DM	4.3	2.8
Total carbohydrates	668	834
NFC	403	195
Carbohydrate fraction, g/kg carbohydrate ^2^		
CA	72	86
CB1	400	16
CB2	132	132
CB3	290	615
CC	112	162
Protein fraction, g/kg CP ^(3)^		
PA+B1	318	467
PB2	548	425
PB3	86	13
PC	49	95

^1^ DM, dry matter; OM, organic matter; CP, crude protein; SOLP, soluble CP; NDICP, neutral detergent insoluble CP; ADICP, acid detergent insoluble CP; aNDF, neutral detergent fiber analyzed using a heat stable amylase and expressed inclusive of residual ash; ADF, acid detergent fiber; ADL, acid detergent lignin; TDN, total digestible nutrients; NEm, net energy for maintenance; NEg, net energy for growth; NFC, non-fiber carbohydrate. ^2^ CA, carbohydrate A fraction, ethanol soluble carbohydrates; CB1, carbohydrate B1 fraction, starch; CB2, carbohydrate B2 fraction, soluble fiber; CB3, carbohydrate B3 fraction, available insoluble fiber; CC, carbohydrate C fraction, unavailable carbohydrate; ^3^ PA+B1, protein A and B1 fractions, soluble CP; PB2, protein B2 fraction, intermediate degradable CP; PB3, protein B3 fraction, slowly degradable fiber-bound CP; PC, protein C fraction, unavailable CP.

**Table 2 animals-10-01270-t002:** Diet composition (g/kg DM or as stated) of the experimental concentrate mix.

Items	Concentrate
Corn, flaked	192
Wheat, ground	99
Corn, ground	8
Lupin, flaked	31
Coconut oil	56
Soybean meal	96
Rapeseed meal	30
Palm kernel meal	71
Corn gluten feed	164
Wheat bran	118
Beet pulp pellet	20
Rice bran	21
Cottonseed hull	9
Limestone	34
Molasses	22
Condensed molasses solubles	11
Salt	8
Sodium bicarbonate	6
Vitamin and mineral mix ^1^	3

^1^ The mix included: 33,330,000 IU/kg vitamin A, 40,000,000 IU/kg vitamin D, 20.86 IU/kg vitamin E, 20 mg/kg Cu, 90 mg/kg Mn, 100 mg/kg Zn, 250 mg/kg Fe, 0.4 mg/kg I and 0.4 mg/kg Se.

**Table 3 animals-10-01270-t003:** BW and feed intake of each measurement period.

	Measurement Period				
Items	Period 1	Period 2	Period 3	SEM	*p*-Value
BW, kg	319	338	373	7.964	<0.001
DMI, kg					
Concentrate	4.6	4.4	5.0	0.012	<0.001
Forage	2.9	2.8	3.0	0.145	0.176
Total	7.4	7.2	8.0	0.151	<0.001

**Table 4 animals-10-01270-t004:** Contribution of variance components affecting BW measurement deviation ^1^.

Factors	Influence, %
Between animals	80.1
Measurement month	17.2
Measurement time within a day	1.9
Measurement day	0.8

^1^ Excluding unexplained random variations. The unexplained error accounted for 70.4% of the total variations.

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
