# Peer review of "Analysis of the Factors Influencing Body Weight Variation in Hanwoo Steers Using an Automated Weighing System"

_animals, 2020, doi:10.3390/ani10081270_

Round 1
Reviewer 1 Report
I think this was a well written paper, see my comments in the attached file.

Author Response
Reviewers' Comments:
#1 reviewer:
I think this was a well written paper, see my comments in the attached file.
[Response] The authors appreciate the encouraging comment by the reviewer.
Minor points:
Line 19-20: We found that differences between animals had the greatest impact on BW deviations in Hanwoo steers.
[Response] We revised the text as suggested.
Line 76: Weight at start of trials? Clarify.
[Response] We added the initial BW to line 75-76 according to reviewer's comments.
Line 111: Feces and urine were removed twice each day. I miss some information about the amounts here, and an assessment of whether feces and urine can cause unexplained random variations (in the discussion). See the comment on line 111 in M&M section. In my opinion, it is important to discuss the large amounts of manure that accumulates on the weight as this can may affect deviations in daily measurements.
[Response] The authors appreciate the reviewer’s comment. The authors agree with the reviewer that the fecal and urinary excretion can cause variations in the BW measurements. As the reviewer pointed out, these variations are hardly explainable errors. We did not include the manure excretion as a response variable but treated as a random error and tried to minimize its possible effects. As the reviewer suggested, we added a discussion of this aspect in the text (line173-177).
Line 140-141: the difference between animals contributed 80.1% of the variation in BW measurements after excluding unexplained random variations, but this is not included as a fixed effect in the M&M section. Is this an explained random effect? Clarify.
[Response] The authors appreciate the reviewer for pointing out this. When we conducted the variance component analysis, we treated all of the factors as random effects. For clarification, we revised the text as follows.
L131-133: The significance of these variance components was analyzed using PROC MIXED of SAS to calculate the level of variation caused by each factor. In this analysis, all of the factors were treated as random effects.
Figure 1: Explain use of the letters a, b and c in the figure.
[Response] The authors appreciate the reviewer’s comment. A description of a, b and c has been added to the legend of Figure 1 according to the reviewer’s comment.
Line 196-201: Contains a discussion of results which should be removed to the discussion section.
[Response] We revised the text as suggested.

Reviewer 2 Report
The availability of fecal output information would have been an added validation to the observations of bw changes.
Author Response
Reviewers' Comments:
#2 reviewer:
The availability of fecal output information would have been an added validation to the observations of BW changes.
[Response] The authors appreciate the reviewer’s comment. The authors agree with the reviewer that the fecal and urinary excretion can cause variations in the BW measurements. However, the authors excluded this factor from the list of response variables because the variations caused by manure excretion are hardly explainable errors. Therefore, we treated the manure excretion as a random error and tried to minimize its possible effects. As the reviewer suggested, we added a discussion of this aspect in the text (line173-177).

Reviewer 3 Report
This paper study the factors that can affect the BW variations in Hanwoo steers by automatic method.
The authors conclude that the difference between individual steers was most influential, and BW measurement deviations were affected by month, day, and the time within a day.
In fact, I did not find any explanation as to why individual variations in the animal affect the BW by AWS.
Is the BW of light steers more accurate? Or the reverse is true.
Are the differences related to animal behavior?
This paper can be published if more data is mentioned about BW of each animal (Initial BW before the beginning of the experiment, during and in the end of the experiment), and if these informations are analyzed in more detail.
In a similar way, the research did not provide any explanation or discussion as to why the BW measurement deviations were affected by month, day, and the time within a day.
As mentioned by the authors; The diet is designed to gain an ADG of 1 kg/d.
Taking into consideration that the authors discussed the results that BW measurement were affected by day and the time within a day, and they explained that feed intake and excretion of feces and urine lead to BW changes during the day. But they did not provide any discussion of why the BW were affected by months. Did the animals gain weight after three months of experience are the reason for that BW measurement deviations were affected by month?
Are the differences due to the design of the AWS device?
Are the differences due to the animal's behavior?
Are the differences due to designing /conditions the experiment?
I think that there is a lack of many necessary details about the experiment design, the way measurements are made and the method of conducting statistical analyzes.
Author Response
Reviewers' Comments:
#3 reviewer:
This paper study the factors that can affect the BW variations in Hanwoo steers by automatic method. The authors conclude that the difference between individual steers was most influential, and BW measurement deviations were affected by month, day, and the time within a day. In fact, I did not find any explanation as to why individual variations in the animal affect the BW by AWS. Is the BW of light steers more accurate? Or the reverse is true. Are the differences related to animal behavior?
[Response] The authors appreciate the reviewer’s comments. The individual variation is always the most significant and influential variable causing variations (deviations) in the Animal Science studies. Each animal has its own characteristics in terms of posture on the weighing scale, steps and movement, and other behavior as the reviewer mentioned. Therefore, it has been commonly assumed that the variation caused by the animal variation would be the major variance component in the BW measurements using AWS. However, its effects have not been quantified, and this manuscript is the first study that presents the relative contribution, as numbers, of the animal factor and others on the BW measurements using AWS. We added a discussion about this more in detail in the text.
This paper can be published if more data is mentioned about BW of each animal (Initial BW before the beginning of the experiment, during and in the end of the experiment), and if these informations are analyzed in more detail.
[Response] The authors appreciate the reviewer’s comments. Since we standardized and quantified BW of each animal by calculating the difference between records of AWS and manual weighing, BW themselves were not of our interest. Nevertheless, the initial mean BW of the animals in each period have been added in Table 3. Feed intakes are also presented in Table 3.
In a similar way, the research did not provide any explanation or discussion as to why the BW measurement deviations were affected by month, day, and the time within a day. As mentioned by the authors; The diet is designed to gain an ADG of 1 kg/d. Taking into consideration that the authors discussed the results that BW measurement were affected by day and the time within a day, and they explained that feed intake and excretion of feces and urine lead to BW changes during the day. But they did not provide any discussion of why the BW were affected by months. Did the animals gain weight after three months of experience are the reason for that BW measurement deviations were affected by month? Are the differences due to the design of the AWS device? Are the differences due to the animal's behavior? Are the differences due to designing /conditions the experiment? I think that there is a lack of many necessary details about the experiment design, the way measurements are made and the method of conducting statistical analyzes.
[Response] The authors appreciate the reviewer’s comments. It was unavoidable that the BW measurement deviations varied by month because the position of the weighing scales might be different, and BW and intake of the animals differed (increased). As suggested, we added a discussion about this in the discussion section. In addition, more detailed descriptions of weight measurement methods and experimental designs were added in the materials and methods section.
